# Fairness Implications of GNN-to-MLP Knowledge Distillation

**Margaret Capetz**
Department of Computer Science
UCLA
mcapetz17@g.ucla.edu

**Yizhou Sun**
Department of Computer Science
UCLA
yzsun@cs.ucla.edu

**Arjun Subramonian**
Department of Computer Science
UCLA
arjunsub@g.ucla.edu

## Abstract

Graph neural networks (GNNs) are increasingly deployed in high-stakes applications where fairness is critical. However, existing data in these real-life scenarios is unreliable, characterized by bias and imbalance. While knowledge distillation (KD) has proven effective to distill GNNs into fully-connected neural networks for scalability, the fairness consequences of such distillation with biased data remain unexplored. Through a systematic evaluation of fairness across synthetic and real-world datasets, we observe that distillation from GNNs to MLPs generally degrades fairness. Our results highlight the need for network-specific considerations when developing mitigation strategies for fairness degradation during knowledge distillation. You can find our code at: https://github.com/mcapetz/fairkd-mlp.

## 1 Introduction

The challenge of developing reliable ML systems from biased data is especially acute in graph learning, in which multiple sources of unreliability accumulate: biased node features, imbalanced class distributions, and connectivity patterns. While graph neural networks (GNNs) have demonstrated high accuracy on node classification tasks, as they exploit graph topology information, this comes with scalability issues. Zhang et al. [2022] have shown that GNNs may be distilled into multilayer perceptrons (MLPs) via knowledge distillation (KD). During KD, a complex teacher model is first trained on data instances and generates soft labels for them. Then, a simpler student model can learn from both the hard and soft labels for instances, often yielding competitive performance with the teacher [Hinton et al., 2015]. KD can enable large-scale industrial machine leaning (ML) applications, as it can drastically reduce model inference time and maintain competitive accuracy. However, we must also consider the reliability of distilled models with regards to their fairness consequences. GNNs are known to be unfair; feature-label associations in training data are often biased, and GNNs inherit and often amplify such biases through graph structure and message passing [Dai and Wang, 2021]. Such unreliability can lead to unfair decision-making, which may further marginalize minoritized and oppressed social groups.

Reliable ML is critical in many fairness-critical real-life scenarios in which the only available data is biased [Mehrabi et al., 2021]. For example, consider the situation in which a bank uses a GNN to make loan decisions. The bank is scaling up operations and considers employing an efficient GNN-to-MLP KD approach to make loan decisions; however, they do not know how distillation affects the fairness of loan decisions. Traditional bank lending has been shown to be discriminatory

39th Conference on Neural Information Processing Systems (NeurIPS 2025) Workshop: Reliable ML from Unreliable Data.

[Tran and Winters, 2024, R. Atkins, 2022], and distilled GNNs may have the potential to further marginalize vulnerable populations *or* alleviate discriminatory decision-making. With this work, we aim to systematically investigate the fairness consequences of GNN-to-MLP KD [Zhang et al., 2022].

Our contributions are the following: We evaluate the fairness outcomes of GNN-to-MLP KD across multiple real-world and synthetic datasets. We conduct a systematic and controlled analysis between teacher versus student models by using different fairness measures based on group accuracy and AUC differences. We further investigate the influence of biased data on the reliability of KD by investigating how graph characteristics impact fairness outcomes in distilled GNNs.

First, we study GNN-to-MLP KD on commonly-used graph fairness datasets and observe that it is difficult to draw concrete conclusions, especially as the existing datasets are biased in terms of class and group balance. Thus, we conduct a closer investigation of KD fairness by studying group performance disparities across synthetic datasets with varying sources of data unreliability (i.e., class, group, and connectivity imbalances), outlined in Figure 1. We find that MLP students generally amplify the unfairness of GNN teachers, demonstrating fairness degradation in GNN-to-MLP KD. In addition, the interdependence of network properties (e.g., group and connectivity imbalances) influence fairness outcomes. Our findings call for KD protocols that are both fairness- and network-aware.

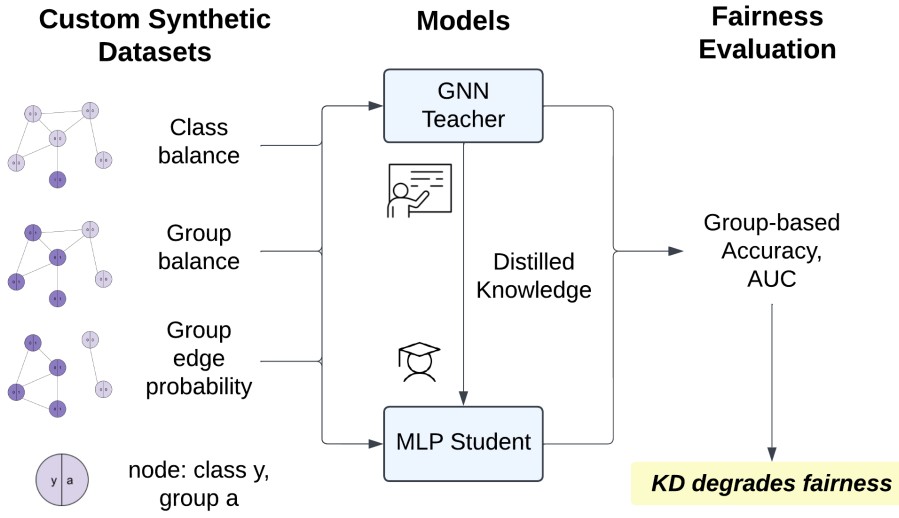

Figure 1: Process of Systematic Evaluation of Fairness. We create a custom synthetic graph dataset with set network properties including class balance, group balance, and group edge probability. A trained GNN teacher is trained on the entire graph to generate soft targets, or teacher logits. The MLP student is trained on the node features guided by the soft targets. The GNN teacher and distilled MLP are deployed for inference and evaluated on fairness metrics. We find that the GNN teacher is generally more fair than the MLP student, with respect to group-based accuracy and AUC fairness measurements, suggesting that KD from GNNs to MLPs can degrade fairness.

## 2   Related Work

**GNN to MLP Knowledge Distillation**   With interest in high-efficiency, high-accuracy models for graph ML, model fairness is critical to evaluate. Zhang et al. [2022] have shown the possibility of distilled GNNs to provide a model with improved efficiency and competitive accuracy, utilizing knowledge distillation originally proposed by Hinton et al. [2015]. Further work has been done in the direction of bridging GNNs and MLPs: Hu et al. [2021] use neighboring contrastive loss to leverage graph structure and provide a supervision signal for the MLP, Yang et al. [2023] propose an intermediate model class that follows an MLP architecture during training and GNN architecture during testing, and Han et al. [2023] use an analog MLP as an initialization step for GNN training.

**GNN Fairness** Recent work has highlighted fairness concerns in graph learning, as graph-based models can lead to unequal outcomes [Laclau et al., 2024]. GNNs are particular susceptible to fairness concerns, as discrimination by GNNs may be magnified by graph topology and message passing [Dai and Wang, 2021]. Fairness issues arise in various graph tasks, including node classification, link prediction, and link classification [Chen et al., 2023a]. In this work, we focus on node classification, specifically how bias in network properties (e.g. class imbalance, group imbalance, and group connectivity) influence fairness in this task. To measure fairness, we consider commonly-used fairness metrics including demographic parity (DP), also known as statistical parity [Zemel et al., 2013], and equal opportunity (EO) [Hardt et al., 2016]. We also consider classification performance metrics including accuracy and Area Under the ROC Curve (AUC), inspired by Qian et al. [2024] and Thölke et al. [2023].

**Fairness of Knowledge Distillation** Recent work has shown that in KD, student models often maintain and even amplify exhibited bias from the teacher model [Lukasik et al., 2021, Chen et al., 2023b]. Some work has begun to address this issue for GNN-to-GNN KD. Dong et al. [2023] have taken initial steps in addressing fairness by adding a general learnable proxy of bias for the student GNN model. Further, [Li et al., 2024] and [Zhu et al., 2023] have explored multi-expert solutions, in which a nodes-only MLP expert and topology-only GNN expert form a composite teacher to produce soft labels for the GNN student. However, fairness implications of GNN-to-MLP KD remain to be studied; in particular, whether the simpler MLP student maintains, amplifies, or reduces the unfairness of the teacher GNN. Our work aims to answer this question empirically.

## 3 Methods

### 3.1 Fairness Metrics

The fairness metrics we first consider are demographic parity and equal opportunity [Zemel et al., 2013, Hardt et al., 2016] Demographic parity (DP) is satisfied if a model's prediction $\hat{y}$ is not dependent on a given sensitive attribute $a$. We operationalize this by measuring the difference between the means of the predicted values conditioned on the sensitive groups. Equal opportunity (EO) is satisfied if both protected and unprotected groups have equal true positive rates, i.e., the probability of correctly identifying a positive case (i.e., $y = 1$) is the same across groups.

$$\text{DP} = |\mathbb{E}[\hat{y}|a = 0] - \mathbb{E}[\hat{y}|a = 1]| \tag{1}$$
$$\text{EO} = |\mathbb{E}[\hat{y}|a = 0, y = 1] - \mathbb{E}[\hat{y}|a = 1, y = 1]| \tag{2}$$

Zhang et al. [2022]'s work on KD from GNN to MLP evaluates models based on accuracy. We propose quantifying fairness by calculating the discrepancy between group-specific accuracy. In particular, to further investigate the consequences for fairness, we consider the difference in accuracy between group 0 and group 1. Note that accuracy is distinct from DP and EO, which do not entirely capture prediction accuracy and focus on equal treatment of groups in terms of positive predictions.

$$\text{Acc Difference} = |\text{Acc}(a = 0) - \text{Acc}(a = 1)| = |\mathbb{E}[1\{\hat{y} = y\}|a = 0] - \mathbb{E}[1\{\hat{y} = y\}|a = 1]| \tag{3}$$
$$\text{AUC Difference} = |\text{AUC}(a = 0) - \text{AUC}(a = 1)| \tag{4}$$

Brzezinski et al. [2024] suggest that traditional fairness metrics like DP and EO are sensitive to data bias including class imbalance, motivating us to consider other metrics to better understand the predictive quality of each group in a more interpretable and reliable manner. We consider accuracy because it was the classification performance metric used in Zhang et al. [2022]. However, accuracy is highly sensitive to class imbalance. Thölke et al. [2023] state that models trained on highly imbalanced data and evaluated based on the accuracy metric yield misleadingly high performances that result from systematically predicting the majority class. The paper suggests using AUC, which plots the true positive against false positive rate for all decision thresholds. Compared to accuracy, AUC is less sensitive to dataset imbalance, making it a better metric to measure fairness implications. Looking at the disparity of group-specific AUC, we can determine if the model is discriminating when classifying for group 0 vs. group 1. Lower values indicate more equitable performance among sensitive groups, while higher values reveal disparities in classification accuracy. By evaluating fairness with AUC disparity, we alleviate the sensitivity of our fairness measurements to biased data.

## 3.2 Existing Datasets

We begin our investigation with common semi-synthetic and real-world datasets in fair graph learning works, listed in Table 1. In particular, we consider semi-synthetic German and Credit datasets and real-world Pokec-z, Pokec-n, and NBA datasets. We supplement this with additional datasets from Qian et al. [2024], including fully synthetic syn-1 and syn-2 datasets and Twitter-based datasets of Sport and Occupation. The heterogeneity in network properties across datasets create confounding factors that made it difficult to isolate if data reliability correlates with KD fairness outcomes. The systematic biases and imbalanced representations across classes and groups make the existing data unsuitable for systematically gauging fairness effects reliably. Thus we explore the consequences of biased data with a systematic approach that utilizes synthetic data in a controlled setting.

## 3.3 Constructing Synthetic Datasets

To understand precisely which factors give rise to unfairness, we utilize a controlled environment for experimentation with synthetic dataset construction. We investigate the relationship between the dataset characteristics and fairness implications based on the work of Sagawa et al. [2020]. We adapt their method for generating toy data with spurious correlations to construct synthetic Stochastic Block Models (SBMs) with varying class balance, group balance, and edge probabilities.

**Block structure.** Following Sagawa et al. [2020], the class label $y \in \{0, 1\}$ is correlated with the sensitive attribute or group label $a \in \{0, 1\}$. We generate graphs following the Stochastic Block Model (SBM) with four blocks representing the combinations of class and group labels: $(y = 0, a = 0)$, $(y = 0, a = 1)$, $(y = 1, a = 0)$, and $(y = 1, a = 1)$. The size of each block is determined by the class balance parameter $c$ that controls the proportion of nodes in class $y = 0$, and the group balance parameter $p$ that controls the proportion of nodes in group $a = 0$. Specifically, for a graph with $N$ nodes, the block sizes are computed as:

$$
\begin{aligned}
|B_{y=0,a=0}| &= N \cdot c \cdot p \\
|B_{y=0,a=1}| &= N \cdot c \cdot (1-p) \\
|B_{y=1,a=0}| &= N \cdot (1-c) \cdot p \\
|B_{y=1,a=1}| &= N \cdot (1-c) \cdot (1-p)
\end{aligned}
\tag{5}
$$

**Edge probability matrix.** The edge probability matrix $P \in \mathbb{R}^{N \times N}$ determines the likelihood of connections between nodes in different blocks. We define a base probability of 0.1 for connections within the same group (i.e., nodes sharing the same value of $a$), and a reduced probability $0.1q$ for connections between different groups, where $q \in [0, 1]$ is the edge probability ratio parameter. This gives us the following edge probability matrix structure $P$, with rows and columns ordered as $[(y = 0, a = 0), (y = 0, a = 1), (y = 1, a = 0), (y = 1, a = 1)]$:

$$
P_{ij} = \begin{cases} 0.1, & \text{if } a_i = a_j \\ 0.1q, & \text{if } a_i \neq a_j, \end{cases}
\tag{6}
$$

where $a_i$ represents the sensitive attribute corresponding to node $i$.

**Node feature generation.** For each node $i$, we generate a feature vector $x_i \in \mathbb{R}^{2d}$, where $d = 100$ is the number of core features. The first $d$ dimensions are the core features correlated with the class label $y_i$, while the remaining $d$ dimensions are spurious features correlated with the group label $a_i$. Thus there are $2d = 200$ total channels. Similar to Sagawa et al. [2020]'s experimentation, we set the number of nodes to 3000, $\sigma_{core} = 10$, and $\sigma_{spu} = 1$. We center class 0 and group 0 at $-1$ and class 1 and group 1 at $+1$ in the core and spurious feature spaces, respectively. Thus, core features correlate with class labels $y$, and spurious features correlate with group labels $a$.

We first initialize all node features as random values drawn from a standard normal distribution: $z_{ij} \sim \mathcal{N}(0, 1)^{2d}$. Note that $z_{ij}$ refers to feature $j$ of node $i$. Next, we modify the core features (first $d$ dimensions) based on class:

$$
x_{ij} = \begin{cases} \sigma_{core} \cdot z_{ij} - 1 & \text{if } j \leq d, y_i = 0 \\ \sigma_{core} \cdot z_{ij} + 1 & \text{if } j \leq d, y_i = 1 \end{cases}
\tag{7}
$$

We modify spurious features (dimension $d$ onwards) based on group:

$$x_{ij} = \begin{cases} \sigma_{spu} \cdot z_{ij} - 1 & \text{if } j > d, a_i = 0 \\ \sigma_{spu} \cdot z_{ij} + 1 & \text{if } j > d, a_i = 1 \end{cases} \tag{8}$$

## 4 Results

**Existing datasets.** To begin, we build upon Zhang et al. [2022]'s codebase and load additional fairness datasets from Qian et al. [2024]: German, Credit, NBA, Syn-1, Syn-2, Sport, and Occupation (Table 1). German, Credit, and NBA are commonly-used fairness datasets. Syn-1 and syn-2 are synthetic datasets developed by Qian et al. [2024]. Sport and Occupation are based on Twitter data. The initial results show that, as expected, the overall accuracy is comparable between the teacher and student models. We consider the traditional fairness metrics DP and EO; these metrics are comparable for most datasets, prompting further investigation.

In Figure 2, we observe the differences in traditional fairness metrics (DP, EO) and classification performance metrics (accuracy and AUC disparity) across existing datasets. German, Credit, and NBA demonstrate smaller differences across all metrics, which is interesting given the different numbers of nodes across datasets. Sport and Occupation datasets, which are based on Twitter data, exhibit high fairness discrepancies compared to other datasets. Syn-1 is constructed with a balanced group ratio, while syn-2 is constructed with an imbalanced group ratio. The different results across syn-1 and syn-2 motivate a systematic investigation of how network properties influence fairness outcomes. Thus, we move from analysis of existing fairness datasets to controlled experimentation with the aforementioned synthetic datasets.

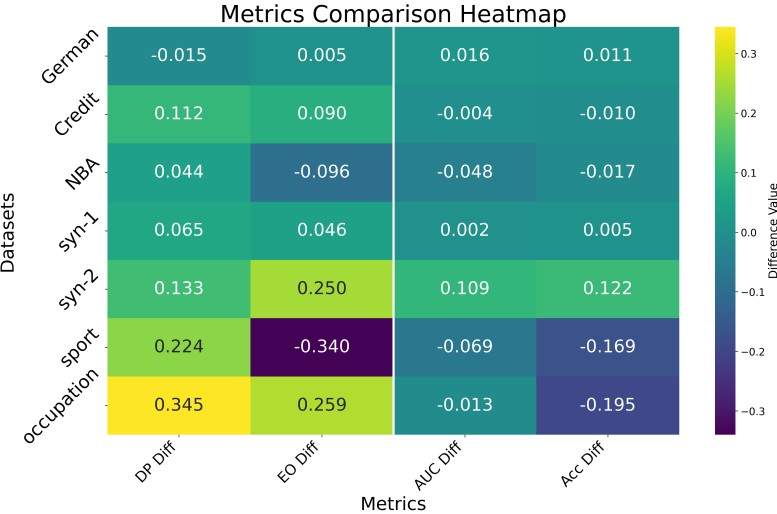

Figure 2: Existing Datasets: Heatmap of Teacher - Student differences for metrics including demographic parity (DP), equal opportunity (EO), AUC, and accuracy. German and syn-1 have generally smaller differences compared to the other datasets.

**Synthetic datasets.** We then conduct a further investigation of the fairness across different network structures with experiments on SBM synthetic datasets. We consider network properties of class balance ($c$), group balance ($p$), and group edge probability ($q$). First, we consider one-factor plots, comparing accuracy and AUC differences with respect to the independent variables $c$, $p$, and $q$. Next, we explore the joint effects of different properties on accuracy and AUC differences. Finally, we investigate if the teacher fairness predicts the student fairness. See more details in Appendix A.

First, we compare accuracy and AUC differences with respect to network properties $c$, $p$, and $q$ in one-factor plots. In the accuracy disparity plots (Figure 3), we see similar trends between the teacher and student, as well as overlapping error intervals. We conclude that one-dimensional analysis does not suggest differences between the teacher and student fairness with respect to any of the network properties. In the AUC plots (Figure 8), the teacher and student demonstrate similar trends, yet there

is a greater difference between values. Since AUC is threshold-agnostic, it may reveal more nuanced fairness trends compared to accuracy.

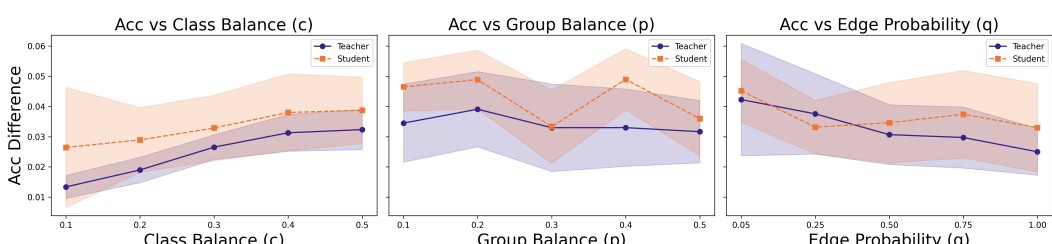

Figure 3: One-factor plots of class balance ($c$), group balance ($p$), and edge probability ($q$) vs. accuracy difference, comparing teacher and student fairness. The teacher generally demonstrates a lower accuracy difference between group 0 and group 1, suggesting better fairness. However, it is important to note that the error intervals overlap.

Next, we perform a two-dimensional analysis by observing the joint effects of network properties on fairness. Across the heatmaps Figure 4 and Figure 9, the teacher is generally fairer than the student, indicating fairness degradation in GNN-to-MLP KD. When comparing accuracy and AUC differences, it seems that AUC reveals more nuanced interactions between network properties and is more sensitive to joint effects. Lower fairness values are located in different regions for different heatmaps, demonstrating the complexity of the joint effects of network properties.

In Figure 4, we see a general trend in which less biased data correlates with less fairness degradation. For the first row comparing class and group balance, we see that the fairness degradation between teacher and student in the most imbalanced case ($c = p = 0.1$) is -0.011 whereas the degradation in the most balanced case ($c = p = 0.5$) is -0.007. However, the error intervals overlap, as seen in Figure 10, suggesting the difference in values is not statistically significant. The two cases of greatest fairness degradation (-0.060) are when $c = 0.1$ and $p = 0.2$ and 0.4, indicating that class balance may have a stronger negative effect on fairness degradation compared to group balance.

In the second row comparing class balance and group edge probability ratio, we observe a more extreme trend of less biased data correlating with less fairness degradation. There is fairness degradation (-0.014) in the most biased case ($c = 0.1$, $q = 0.05$) in which the classes are most imbalanced and the groups are connected most sparsely. Moreover, there is an improvement in fairness (+0.005) for the least biased case ($c = 0.5$, $q = 1$) in which the classes are most balanced and the groups are connected most densely. The error bars (Figure 10) do not overlap, suggesting possible statistical significance. In the accuracy difference heatmap, we see that $q$ may have a stronger influence on fairness degradation than $c$ and that fairness degradation gets low as $q$ approaches 1, in relation to $c$. Across the three rows, the middle row has values closest to zero in terms of teacher and student fairness disparity, demonstrating that $c$ and $q$ may have a more stable joint effect on fairness.

In the final row, we see that there is actually a fairness improvement in what would be considered the most biased case of greatest group imbalance and group connection sparsity ($p = 0.1$, $q = 0.05$) of 0.017 and a fairness degradation in the least biased case of greatest group balance and group connection density ($p = 0.5$, $q = 1$). The error bars (Figure 10) overlap, suggesting the difference is values is not statistically significant. The hot spots of greatest fairness improvement and greatest fairness degradation are at the same group balance setting, $p = 0.2$, and at $q = 0.25$ and $q = 1$ respectively. Contrary to the previous row in which fairness degradation generally gets low as $q$ approaches 1 in relation to $c$, in this row we observe that fairness degradation generally increases as $q$ approaches 1 in relation to $p$.

However, it is important to consider the possible overlaps of error intervals as seen in Figure 10 and Figure 11 to understand the statistical significance of results. A greater number of seeds should be run to more rigorously analyze the statistical significance of the results.

Next, we analyze whether teacher fairness values predict student fairness values by plotting scatterplots of teacher fairness vs. student fairness. In Figure 5 and Figure 6, the $R^2$ values are 0.324 and 0.277, respectively, indicating a low/moderate correlation between teacher and student values. The

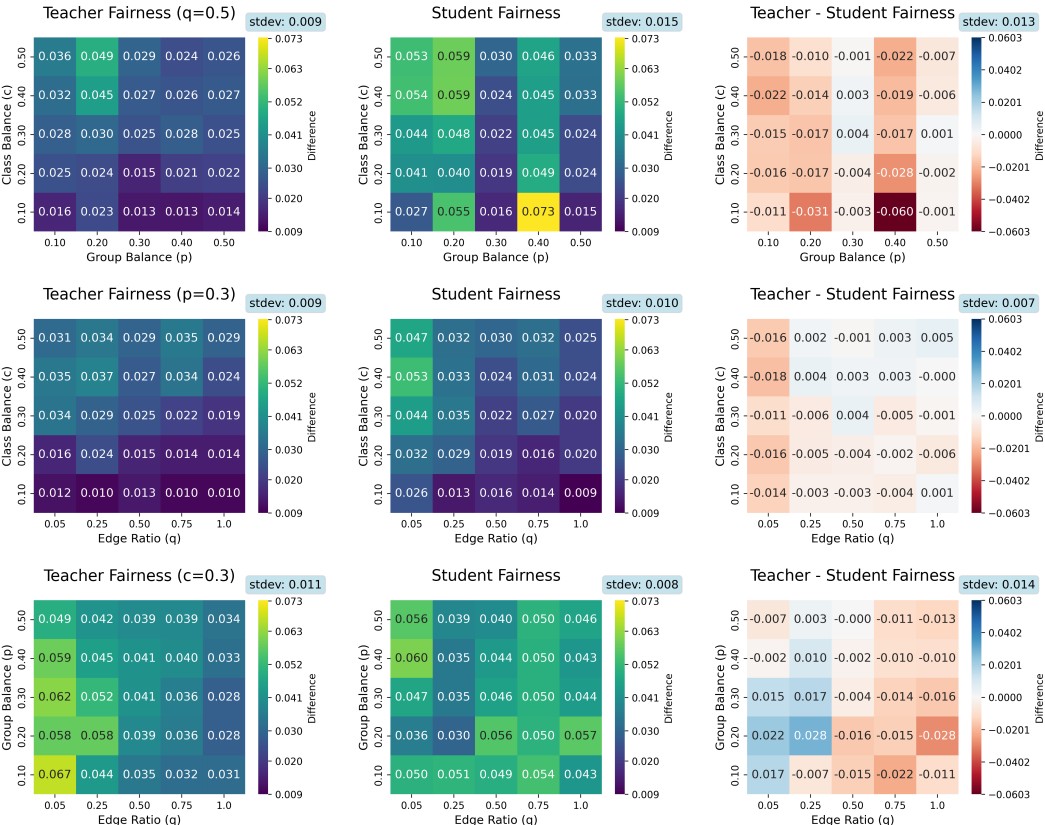

Figure 4: Two-factor heatmaps of class balance ($c$), group balance ($p$), and edge probability ($q$) vs. accuracy difference, comparing teacher and student fairness. Looking at the final column of the difference between teacher and student, most values are red (negative), which suggests that the teacher has better fairness than the student. The standard deviation is over the values in the heatmap. See Figure 10 for the heatmap with error bars.

low p-values indicate high statistical significance. The AUC scatterplot has more values above the $y = x$ line, suggesting that the student is less fair than the teacher. To further investigate the effects of network properties, we colored the points based on the parameters $c, p, q$ to observe clustering. The most clear trend was evident in the group balance ($p$) colorings for the accuracy disparity metric, as seen in Figure 7. This may suggest the importance of $p$ in comparison to other network properties.

## 5 Discussion

Our work aims to understand the fairness implications of KD from GNNs to MLPs. In our initial experiments with existing datasets and traditional fairness metrics, we were faced with inconclusive results due to heterogeneity of dataset network properties being a possible confounding factor. The most comparable existing datasets are the two synthetic datasets generated by Qian et al. [2024]. Syn-1 has group balance, while syn-2 has group imbalance. As seen in Figure 2, the resulting measurements DP, EO, AUC difference, and Acc difference are very different for syn-1 and syn-2. When observing the confusion matrices across datasets, we saw that the existing fairness datasets are imbalanced with respect to class and group. The models were achieving high overall accuracy by solely predicting the majority class. By evaluating in terms of group-based accuracy disparity instead of overall accuracy, we can see whether the model is disproportionately failing to make correct predictions for the minority group. In addition, we consider group-based AUC which is less sensitive to class imbalance. Thus, the takeaways of our initial experiments on existing datasets are that (1) existing real-world and synthetic fairness datasets are insufficient for systematically evaluating the

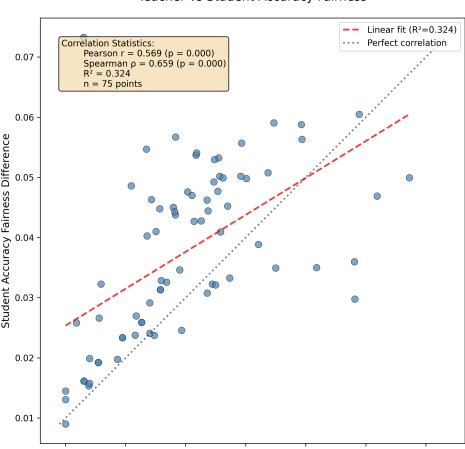

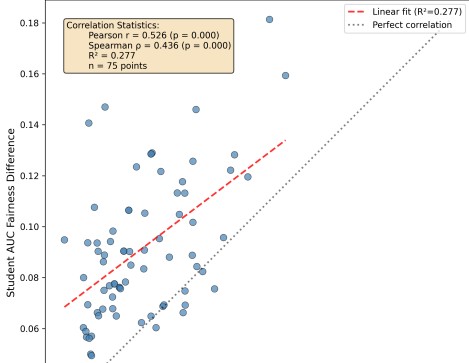

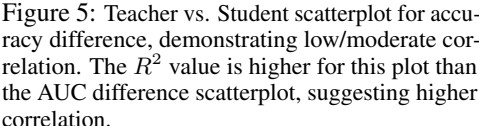

Figure 5: Teacher vs. Student scatterplot for accuracy difference, demonstrating low/moderate correlation. The $R^2$ value is higher for this plot than the AUC difference scatterplot, suggesting higher correlation.

Figure 6: Teacher vs. Student scatterplot for AUC difference, demonstrating low/moderate correlation. Most points are above the $y = x$ line, suggesting that the student model is less fair compared to the teacher model when considering AUC difference.

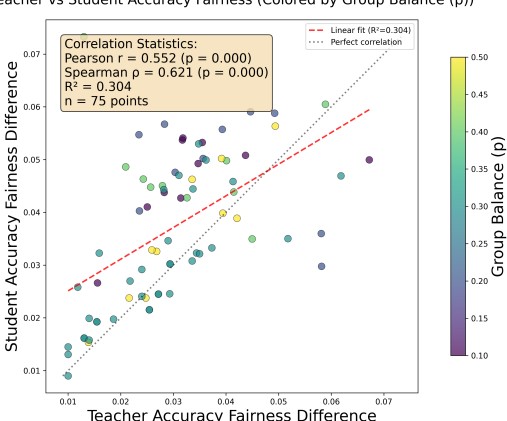

Figure 7: Teacher vs. Student scatterplot for accuracy difference, colored by group balance ($p$). The yellower points are more balanced and closer to the linear fit and perfect correlation lines, indicating higher correlation when groups are more balanced. The darker purple points are almost all above the lines, indicating that the student is almost always less fair than the teacher when the dataset has greater group imbalance.

effects of network properties on the fairness outcomes of KD and (2) traditional fairness metrics (DP, EO, Acc difference) are sensitive to dataset bias.

Thus, we propose a systematic and controlled framework to test the influence of network properties on fairness. We use a fairness measurement that is less sensitive to biased data: AUC disparity. To control for dataset bias, we experiment with SBMs with varied class balance, group balance, and group edge probability. In our one-factor analysis of fairness trends (Figure 3, 8), we find that the one-dimensional analysis does not yield any conclusive trends. In our two-dimensional analysis, we see that the teacher is generally fairer than the student, thus we come to the conclusion that GNN-to-MLP KD generally degrades fairness. This aligns with previous conclusions about KD from GNNs to GNNs [Lukasik et al., 2021, Li et al., 2024, Zhu et al., 2023]. Fairness degradation from teacher to student is evident in both group-based Acc and AUC heatmaps (Figure 4, 9): the heatmaps in the rightmost column portraying teacher–student differences are mostly red, indicating that the teacher is fairer than the student. Additionally, we demonstrate some potential trends based on the

heatmap trends. Compared to group balance and group edge probability, class balance may have a more prominent influence on group-based Acc differences, but less of an impact on group-based AUC differences. We also observe that while the teacher and student behavior varies in trends across different heatmaps, there is less teacher-student variance in the heatmap that varies class balance and group edge probability, possibly suggesting there is more stability in this particular joint effect on group-based Acc differences. However, this trend is not evident in the group-based AUC difference heatmaps. Although we can clearly see that KD generally degrades fairness across group-based Acc and AUC, it is difficult to come to more nuanced concrete conclusions about network property effects with our results. Broader experimentation across larger datasets, more network properties, and different graph tasks is necessary to come to more solid conclusions.

Finally, we analyze whether teacher fairness predicts student fairness. The correlation coefficients are 0.324 and 0.277 for Acc difference and AUC difference respectively, demonstrating a low/moderately predictive relationship between the teacher and student. This experiment confirms that teacher fairness serves as a meaningful predictor of student fairness. In addition, many points are above the equality line, demonstrating that the student generally has worse fairness than the teacher. We observe more degradation for group-based AUC differences than group-based Acc differences (Figure 5, 6). The scatterplot results reinforce the previous conclusion that GNN-to-MLP KD degrades fairness. In addition, coloring the scatterplot by group balance (Figure 7) reveals student fairness degrades more when groups are more imbalanced, reinforcing the notion that more biased data leads to greater fairness degradation. This demonstrates a trend not clear in the previous heatmaps, yet this does not offer a concrete conclusion about the joint effects of network properties on fairness.

Our work motivates KD fairness interventions to take into account network properties. When deploying student models, special attention should be paid to scenarios with biased data, such as extreme dataset imbalance in regards to class or group ratio, as well as extremely sparse or dense group connectivity. In addition, modifications to KD techniques may be needed to better preserve fairness properties for more reliable ML.

In future work, the impact of network properties on fairness can be further explored by extending upon SBMs to investigate more complex network structures (hierarchical networks, multilayer networks) and more nuances in biased data (intersectional discrimination). In addition, fairness-aware distillation methods can be investigated, including incorporating fairness constraints or regularization terms to prevent fairness degradation. Multi-teacher distillation and staged distillation may also be considered [Li et al., 2024, Zhu et al., 2023]. Regarding fairness, future work can also examine intersectional fairness implications, simultaneously looking at fairness across multiple sensitive attributes.

## 6 Conclusion

Through systematically evaluating the fairness of KD across existing and synthetic datasets, and analyzing fairness measurements that are less sensitive to biased data, we unveil the impacts of KD from GNNs to MLPs on fairness. Our results demonstrate an overall fairness degradation from teacher GNN to student MLP when performing KD on biased graph data. The bias and imbalance of network properties have complex joint effects on fairness outcomes. This highlights not only the need for reliable fairness-aware KD protocols but also the need for network-aware fairness interventions. As GNNs are increasingly deployed in fairness-critical applications, we must consider network and fairness-aware KD protocols. Our fairness-focused analysis of GNN-to-MLP KD fits into the broader problem of building reliable ML, demonstrating that efficiency gains may come at the cost of fairness, especially when data quality is compromised.

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

# A   Appendix

Table 1: Statistics of selected datasets used in fair graph learning. Note that all numbers are taken from Qian et al. [2024].

| Dataset | German | Credit | NBA | Syn-1 | Syn-2 | Sport | Occupation |
|---|---|---|---|---|---|---|---|
| # Nodes | 1,000 | 30,000 | 403 | 5,000 | 5,000 | 3,508 | 6,951 |
| # Edges | 21,742 | 1,421,858 | 10,621 | 34,363 | 44,949 | 136,427 | 44,166 |
| # Features | 27 | 13 | 39 | 48 | 48 | 768 | 768 |
| Sensitive Attribute | Gender | Age | Nationality | 0/1 | 0/1 | Race (White/Black) | Gender (Male/Female) |
| Label | Good/bad Credit | Payment default/no default | Salary | 0/1 | 0/1 | NBA/MLB | Psy/CS |
| Average Degree | 44.48 | 95.79 | 53.71 | 13.75 | 17.98 | 78.78 | 13.71 |

**Experimental settings.**    Following Zhang et al. [2022], we use a GCN [Kipf and Welling, 2017] teacher, MLP student, and transductive experimental setting. We run each experiment five times with different random seeds and report the average and standard deviation. All experiments are run with PyTorch [Paszke et al., 2019] and PyTorch Geometric [Fey and Lenssen, 2019] on a machine with 48 Intel(R) Xeon(R) E5-2650 v4 @ 2.20GHz CPUs and 7 NVIDIA Titan XP Graphic Cards each with 12288MiB of space. The total memory used during experimentation is 7.2GiB.

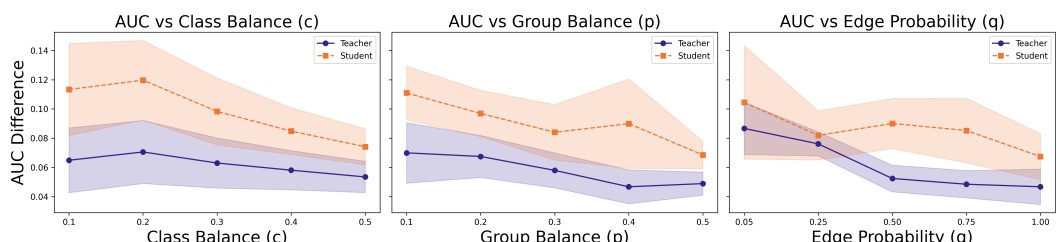

Figure 8: One-factor plots of class balance ($c$), group balance ($p$), and edge probability ($q$) vs. AUC difference, comparing teacher and student performance. The teacher generally demonstrates a lower AUC difference between group 0 and group 1, with less overlapping error compared to accuracy difference, suggesting better fairness.

**Limitations**    In our experimentation, we focus on graph convolutional networks (GCNs), without experimenting on additional GNN architectures like graph attention networks (GATs) and Graph-SAGE. We also focus soley on node classification, and do not consider additional graph tasks like link prediction. The synthetic SBM data we build with a simple four-block structure does not capture heterophily, multi-scale structure, and other complex network properties of real-world graphs. In addition, the fairness analysis is limited to binary sensitive attributes, binary classification tasks, and single sensitive axis analysis. Future work may also analyze fairness outcomes with various alternative KD techniques.

**Broader Impacts**    Our paper aims to improve justice in graph learning. In particular, our analysis of fairness outcomes seeks to inform fairness-aware approaches to GNN-to-MLP KD. Our work has applications in fairness-critical real-life scenarios including bank loan decisions, credit rating, fraud detection, and more.

**LLM Usage.**    LLMs were used for debugging and writing help. They were not used to generate ideas, carry out experiments, or draw conclusions.

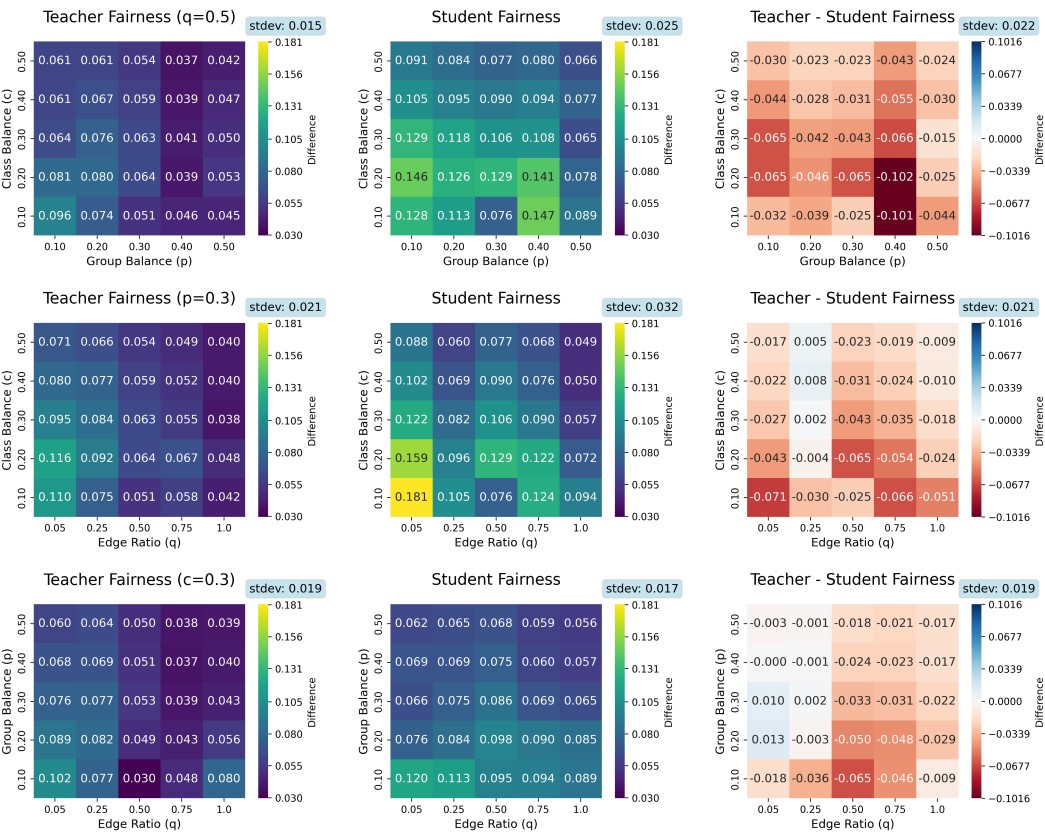

Figure 9: Two-factor heatmaps of class balance (c), group balance (p), and edge probability (q) vs. AUC difference, comparing teacher and student fairness. Class balance seems to have a less prominent effect on fairness compared to group balance and edge ratio. For instance, as class balance and edge ratio increase, the difference in AUC between groups seems to decrease. For most AUC heatmaps, the lowest values seems to be in the upper right corner. Similar to the accuracy difference heatmaps, the student standard deviation is greater than the teacher for the first two rows. Additionally, the range of the colorbar is greater for the AUC heatmaps compared to the accuracy difference heatmaps, demonstrating that the range of AUC disparity is greater than the range of Acc disparity. This is interesting because AUC is less sensitive to dataset imbalance. See Figure 11 for the heatmap with error bars.


Figure 10: Two-factor heatmaps of class balance (c), group balance (p), and edge probability (q) vs. Accuracy difference (with error), comparing teacher and student fairness. These plots include the mean and standard deviation that is calculated across the runs on five seeds.

Answer: [NA]

Justification: The paper does not include theoretical results.

4. **Experimental result reproducibility**

Question: Does the paper fully disclose all the information needed to reproduce the main experimental results of the paper to the extent that it affects the main claims and/or conclusions of the paper (regardless of whether the code and data are provided or not)?

Answer: [Yes]

Justification: The experimental methods are detailed. Code will be made accessible.

5. **Open access to data and code**

Question: Does the paper provide open access to the data and code, with sufficient instructions to faithfully reproduce the main experimental results, as described in supplemental material?

Answer: [Yes]

Justification: The code and data will be made accessible.

6. **Experimental setting/details**

Question: Does the paper specify all the training and test details (e.g., data splits, hyperparameters, how they were chosen, type of optimizer, etc.) necessary to understand the results?

Answer: [Yes]

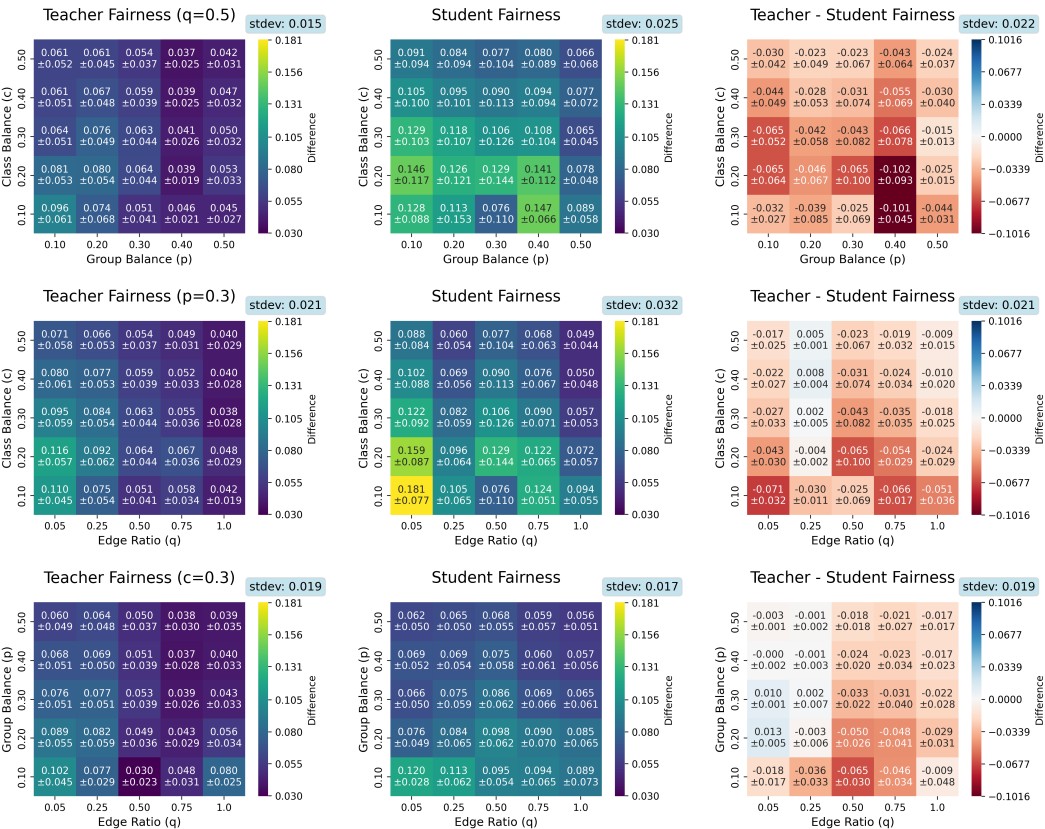

Figure 11: Two-factor heatmaps of class balance (c), group balance (p), and edge probability (q) vs. AUC difference (with error), comparing teacher and student fairness. These plots include the mean and standard deviation that is calculated across the runs on five seeds.

Justification: Experimental settings are detailed in the appendix. See Appendix A.

7. **Experiment statistical significance**

   Question: Does the paper report error bars suitably and correctly defined or other appropriate information about the statistical significance of the experiments?

   Answer: [Yes]

   Justification: Error bars and statistical significance are included in most results. For the heatmaps, the error bars are included in plots in the appendix (Figure 10 and Figure 11) and omitted from the main plots for readability.

8. **Experiments compute resources**

   Question: For each experiment, does the paper provide sufficient information on the computer resources (type of compute workers, memory, time of execution) needed to reproduce the experiments?

   Answer: [Yes]

   Justification: The experimental compute resources are detailed in the appendix. See Appendix A.

9. **Code of ethics**

   Question: Does the research conducted in the paper conform, in every respect, with the NeurIPS Code of Ethics https://neurips.cc/public/EthicsGuidelines?

   Answer: [Yes]

   Justification: The research conforms with the NeurIPS Code of Ethics.

10. **Broader impacts**

    Question: Does the paper discuss both potential positive societal impacts and negative societal impacts of the work performed?

    Answer: [Yes]

    Justification: The paper discusses the societal impacts of the research. See Appendix A.

11. **Safeguards**

    Question: Does the paper describe safeguards that have been put in place for responsible release of data or models that have a high risk for misuse (e.g., pretrained language models, image generators, or scraped datasets)?

    Answer: [NA]

    Justification: The paper poses no such risks.

12. **Licenses for existing assets**

    Question: Are the creators or original owners of assets (e.g., code, data, models), used in the paper, properly credited and are the license and terms of use explicitly mentioned and properly respected?

    Answer: [Yes]

    Justification: All assets used are properly cited. See Section 3.2.

13. **New assets**

    Question: Are new assets introduced in the paper well documented and is the documentation provided alongside the assets?

    Answer: [Yes]

    Justification: We release our code via Github under an MIT license. We provide a README that explains how to reproduce our experiments.

14. **Crowdsourcing and research with human subjects**

    Question: For crowdsourcing experiments and research with human subjects, does the paper include the full text of instructions given to participants and screenshots, if applicable, as well as details about compensation (if any)?

    Answer: [NA]

    Justification: This paper does not involve crowdsourcing nor research with human subjects.

15. **Institutional review board (IRB) approvals or equivalent for research with human subjects**

    Question: Does the paper describe potential risks incurred by study participants, whether such risks were disclosed to the subjects, and whether Institutional Review Board (IRB) approvals (or an equivalent approval/review based on the requirements of your country or institution) were obtained?

    Answer: [NA]

    Justification: The paper does not involve crowdsourcing nor research with human subjects.

16. **Declaration of LLM usage**

    Question: Does the paper describe the usage of LLMs if it is an important, original, or non-standard component of the core methods in this research? Note that if the LLM is used only for writing, editing, or formatting purposes and does not impact the core methodology, scientific rigorousness, or originality of the research, declaration is not required.

    Answer: [Yes]

    Justification: This is detailed in the appendix. See Appendix A.

