# OpenReview forum: "Fairness Implications of GNN-to-MLP Knowledge Distillation"
_NeurIPS.cc/2025/Workshop/Reliable_ML — NeurIPS 2025 - Reliable ML Workshop_

### Official Review · Reviewer_p6ZA · 2025-09-18
**The paper is novel, clear and relevant to reliability with imperfect data. Clear accept.**

**Rating:** 8
**Confidence:** 3

**Review:**

The authors of the paper study the fairness implications of performing knowledge distillation (KD) to distill graph neural networks (GNNs) into multilayer perceptrons (MLPs). They evaluate the fairness outcomes of GNN-to-MLP KD across multiple real-world and synthetic datasets. They conduct a systematic and controlled analysis between teacher versus student models by using different fairness measures based on group accuracy and Area Under the ROC Curve (AUC) differences. They also investigate the influence of biased data on the reliability of KD by investigating how graph characteristics impact fairness outcomes in distilled GNNs. They claim that MLP students generally amplify the unfairness of GNN teachers, demonstrating fairness degradation in GNN-to-MLP KD when the graph data is biased. Specifically, they claim that existing real-world datasets are insufficient for systematically evaluating the effects of network properties on the fairness outcomes of KD and that traditional fairness metrics (demographic parity, equal opportunity, accuracy difference) are sensitive to dataset bias. They also claim that AUC disparity is less sensitive to biased data. Their two-factor analysis (based on network properties like class balance group balance and edge probability) showed that the teacher is generally fairer than the student, implying that GNN-to-MLP KD generally degrades fairness.

The paper appears to be novel, is relevant to reliability with imperfect data and the arguments are presented with clarity. Also, the discussion of the results was highly detailed. No weaknesses observed.

---

### Official Review · Reviewer_mxfA · 2025-09-20
**Thorough work in looking at GNN to MLP KD**

**Rating:** 8
**Confidence:** 4

**Review:**

This paper explores fairness issues that arrive when distilling knowledge from GNNs to MLPs for scalability. KD has historically been used to take powerful ML models (such as GNNs) and make them applicable to a variety of settings with less computational power. While prior work has shown that there is little change in accuracy, this work explores the fairness complications of KD.

Pros:
- Paper evaluates on commonly used fairness metrics and demonstrates a good understanding of the space in related work.
- Several datasets are used that span synthetic and real-world.
- Results are interesting. They support prior work in many places where AUC difference is small but there exist large DP and EO differences in some datasets. While results on different datasets are mixed, the findings are still interesting.
- Plots are carefully chosen and highlight interesting findings that showcase different aspects of their results. I particularly like Figure 3 which connects their results to network properties.

Cons:
- Remove the smiley faces from Figure 1. They're distracting and their placement doesn't add anything.
- The dataset labels in Figure 2 are way too small.
- It would be helpful to break up the writing into smaller subsections and explain the plots individually that way. As a block of text, it's hard to follow and flip back and forth. If space was a problem for this, I think you could cut some of these figures. For instance, I don't think the first two columns in Figure 4 are necessary to demonstrate that teacher is more fair.
- Results are mixed. While I'm convinced there are fairness differences and that 'teacher' is more broadly fair, it's clear that this isn't universal and is also not significant in most settings. However, I appreciate that this nuance was also understood and explained by the authors.